# Unmet care needs of children with ADHD

Richard Vijverberg[1,2,3]*, Robert Ferdinand[1], Aartjan Beekman[3], Berno van Meijel[2,3,4]

**1** GGZ Delfland, Department of Child and Adolescent Psychiatry, Delft, The Netherlands, **2** Inholland University of Applied Sciences, Amsterdam, the Netherlands, **3** Amsterdam UMC, VU Amsterdam, Department of Psychiatry, Amsterdam Public Health Research Institute, Amsterdam, the Netherlands, **4** Parnassia Psychiatric Institute, The Hague, the Netherlands

* r.vijverberg@ggz-delfland.nl

**Data Availability Statement:** Relevant data are available from Figshare (DOI: 10.6084/m9.figshare.11417070).

**Funding:** This research was funded by Delfland psychiatric institute, which played no role in the design of the study, in the collection, analysis or

## Abstract

### Background

Non-compliance to, or drop-out from treatment for childhood ADHD, result in suboptimal outcome. Non-compliance and drop-out may be due to mismatches between patients' care needs and treatments provided. This study investigated unmet care needs in ADHD patients. Unmet needs were assessed in two different treatment settings (general outpatient setting versus youth-ACT). Youth-ACT treatment is an intensive outreach-oriented treatment for patients with severe psychiatric and psychosocial problems. Comparison of a general outpatient sample with a youth-ACT sample enabled us to assess the influence of severity of psychiatric and psychosocial problems on perceived care needs.

### Methods

Self-reported unmet care needs were assessed among 105 ADHD patients between 6 and 17 years of age in a general outpatient (n = 52) and a youth-ACT setting (n = 53).

### Results

ADHD patients most frequently reported unmet needs regarding mental health problems, information on diagnosis/treatment, and future prospects. Outpatients differed from youth-ACT patients with respect to 30% of the unmet care needs that were investigated. Outpatients perceived more unmet needs regarding information on diagnosis/treatment (p = 0.014). Youth-ACT patients perceived more unmet needs concerning medication side effects (p = 0.038), quality and/or quantity of food (p = 0.016), self-care abilities (p = 0.016), regular/suitable school or other daytime activities (p = 0.013), making and/or keeping friends (p = 0.049), and future prospects (p = 0.045).

### Conclusions

Focusing treatment of ADHD patients on unmet needs may reduce non-compliance and drop-out. In clinical practice, systematic assessment of unmet care needs in all ADHD patients may be warranted, e.g. using the CANSAS questionnaire during the screening/intake phase.

interpretation of the data; or in writing the manuscript. Grant number is not applicable.

**Competing interests:** The authors have declared that no competing interests exist.

**Abbreviations:** ACT, Assertive Community Treatment; ADHD, Attention Deficit Hyperactivity Disorder; ASD, Autism Spectrum Disorder; CANSAS, Camberwell Assessment of Need Short Appraisal Schedule; Df, degrees of freedom; FE, Fisher's Exact test; ICF, International Classification of Functioning and Disability; MINI-KID, MINI International Neuropsychiatric Interview for Children and Adolescents; SD, Standard Deviation; SPSS, Statistical Package for the Social Sciences; χ² –test, Chi-square test.

# Background

With a worldwide prevalence rate of approximately 5%, attention-deficit/hyperactivity disorder (ADHD) is one of the most common psychiatric disorders in children and adolescents [1]. ADHD is characterized by excessive and developmentally inappropriate symptoms of inattention- disorganization, hyperactivity and impulsiveness [2]. The consequences of ADHD to well-being and daily functioning vary according to the severity of symptoms and impairments that affect daily activities such as self-care and handling money [3, 4]. They also affect participation in the community (e.g., school attendance and keeping friends) [5, 6].

If intensive psychiatric treatment is needed, children and adolescents with ADHD in the Netherlands are referred to specialized general outpatient clinics by a general practitioner [7]. Treatment generally focusses on reducing symptoms and improving psychosocial functioning [6, 8]. Common treatments include medication (e.g. stimulants); behavioural therapy and cognitive behavioural therapy; psycho-education, organization and planning-skills training; social skills training; and parental support [9, 10]. If even more intensive mental health care is necessary, patients can be referred to youth Assertive Community Treatment (youth-ACT). ACT is an intensive and outreach-oriented treatment for patients with severe psychiatric and psychosocial problems. Treatment is provided by a multidisciplinary team of mental health care professionals [11–13].

Although effective treatments are available, many children and adolescents with psychiatric disorders remain undertreated [14–16]: in over 40% of patients, the proper delivery of psychiatric treatment interventions is hampered by non-attendance, non-compliance, or drop-out [17–22]. Several factors are associated with these problems, one of the most prominent being the mismatch between a patient's perceived care needs and the treatment that is actually provided [15, 23]. Perceived needs can be subdivided into (1) met needs, i.e., difficulties in a particular domain of functioning that are adequately taken care of; and (2) unmet needs, i.e., those for which a patient believes that he or she is not receiving the right care or the appropriate level of care [24].

At present there is little information on the perceptions of children and adolescents with ADHD regarding their met and unmet care needs. To examine these patients' care needs, previous studies have either used small samples [25, 26], or samples that also included young adults with autism spectrum disorder [27]. Other studies on ADHD patients' care needs focused on the parents' perspectives [26, 28]. But while information on the latter is important, it is not enough, especially as a parent's perspective on an adolescent patient may differ significantly from that of the patient himself or herself [24, 29, 30]. Moreover, insight into the perception of ADHD patients may help to enhance their adherence to treatment [31].

To improve our understanding of receiving treatment in specialized mental health care, we studied met and unmet care needs according to the categories of the International Classification of Functioning and Disability (IFC) [6, 32]. Because care needs may vary depending on the intensity of care we included patients who had been referred to an outpatient mental healthcare setting, or to a setting providing youth Assertive Community Treatment (youth-ACT) [11]. This comparison enabled us to judge the influence of severity of psychiatric and psychosocial problems on unmet care needs [13]. Further, since parents are involved differently in younger versus older children, we investigated unmet care needs in two age groups: primary school children and adolescents.

On the basis of the literature, we had two *a priori* hypotheses: (1) that ADHD patients treated in the youth-ACT setting would experience more unmet care needs than those treated in a general outpatient care setting [11, 13, 33]; and (2) that the greatest differences between

patients in the two settings would involve participation in the community, with more recipients of youth-ACT perceiving that their care needs were not being met [2, 34, 35].

## Methods

### Design

This cross-sectional study was conducted between 2015 and 2017 with patients treated in two general outpatient clinics or a youth-ACT setting, all being part of a large mental health care institution in the Netherlands.

### Setting

Participants were recruited from two general outpatient treatment settings and one youth-ACT team.

Treatment in the general outpatient settings was provided by a multidisciplinary team consisting of one child psychiatrist, six psychologists, and one nurse practitioner.

The youth-ACT team consisted of one child psychiatrist, five psychologists, three nurse practitioners and two mental health nurses. It offered outreach-oriented (home-based) treatment to patients with more severe psychiatric and psychosocial problems who were often difficult to reach. Staff in this youth-ACT team had small shared caseloads (<15 patients) and provided intensive and outreach-oriented case management, early intervention, behavioural therapy (including cognitive behavioural therapy), family support, and pharmacological treatment. The intensity of the treatment could be scaled up or down according to the severity of current psychiatric symptoms and a patient's specific psychosocial impairments.

### Participants

Participants were patients aged between 6 and 17 years, all of whom had been diagnosed with ADHD. One child per household was allowed to participate in the study. A random sample was selected from the general outpatient population. For the youth-ACT sample, we included all patients who were referred to this treatment setting during the inclusion period. These ACT-patients all had received prior general outpatient treatment. A total of 121 patients were eligible for inclusion. The final sample consisted of 105 patients: 52 in the outpatient sample and 53 in the youth-ACT sample. Fig 1 presents the flowchart for inclusion.

### Ethical approval

The study was reviewed and approved by the Medical Ethical Committee and the Scientific Committee at the EMGO⁺ Institute for Health and Care Research at VU University Medical Centre in Amsterdam (protocol no. 2015.245), and by the local scientific review board at the participating institution. Written and oral information on the research project was provided separately to the children or adolescent participants and their parents.

In keeping with prevailing legislation in the Netherlands, written consent from the parents and/or children/adolescents was obtained according to three age categories. Category 1: Parents were asked for consent for children younger than 12 years old. Category 2: If children were aged between 12 and 16, parents and children were both asked for consent. Category 3: Informed consent was obtained from an adolescent if he or she was aged 16 or older.

### Measurement instruments

A child or adolescent's demographic characteristics were measured using the Demographic Information Questionnaire (DEMOG), a designated client-based standardized questionnaire

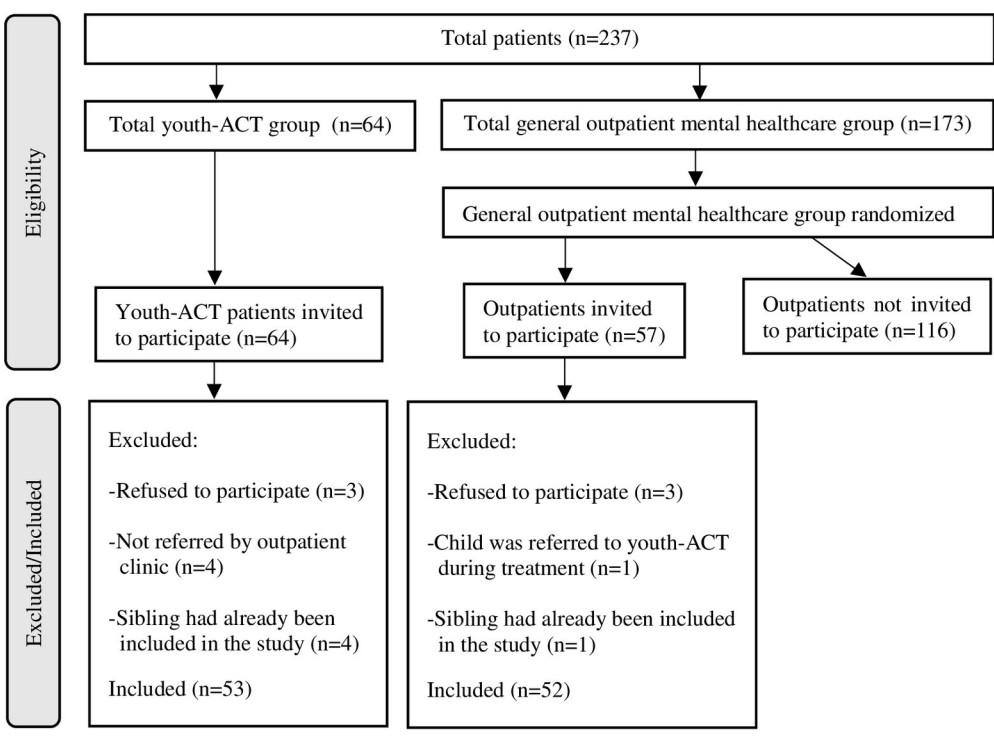

**Fig 1. Participant flow diagram.**

used to measure the following demographic characteristics: (1) age, (2) gender, and (3) living situation [36].

Patients' psychiatric diagnoses were assessed using the Neuropsychiatric Interview for Children and Adolescent (MINI-KID), supplemented with clinical diagnoses based on the DSM-5 that are not included in the MINI-KID [2, 37].

Currently, there is no "gold-standard" for assessing care needs in patients with childhood ADHD. To assess unmet care needs in children and adolescents, the Camberwell Assessment of Need Short Appraisal Schedule (CANSAS) [38] has been used in previous research [27]. The CANSAS was judged as the most appropriate of the available needs assessment instruments, as it is the most widely used needs assessment tool in general mental health services [27]. The CANSAS covers 24 items, each of which distinguishes three levels of care need: (1) no need (= no problem), (2) met need (no or moderate problem because of help received), (3) and unmet need (current serious problem, regardless of any help received) [39, 40]. The CANSAS items were categorized using the following ICF (International Classification of Functioning, Disability and Health) health and health-related domains: (a) physical and mental functions, (b) performance of daily activities, (c) participation in the community [6, 32, 41]. The CANSAS was administered in a face-to-face interview with the patient during the intake procedure.

At the outpatient clinics, measurements were conducted on the day of the first appointment (intake). In the youth-ACT setting, measurements after the first (intake) or second appointment. In both settings, measurements for this study took place before patients and parents were informed about results of the clinical assessments. For children below the age of 12, the interview was carried out in the presence of the parent. The parent was encouraged to support the child in answering the question if the interviewer felt that the child's answer was unclear.

Prior to the interview, parents were instructed not to answer for the child, but to clarify the questions in such a way that the child was able to answer the question from his or her own perspective.

## Data-analysis

To analyse the background characteristics and the total number of self-reported unmet care needs, descriptive statistics were computed, for the overall sample and for the two subgroups separately (patients from the general outpatient setting and patients from the youth-ACT sample).

Subgroup differences were analysed using the *t*-test for continuous variables, chi-square test with Yates continuity correction (= $\chi^2$ -test), or, if the expected number in at least one of the cells was smaller than 5, with the Fisher Exact test [42]. A value below p < 0.05 was considered to be statistically significant.

To investigate the association between age and unmet care needs, we constructed two subgroups, i.e., primary school children (age 6–12 years) and adolescents (age 13–17 years). For the overall sample, we performed the Chi-square test with Yates continuity correction (= $\chi^2$ -test) to analyse differences between these two age groups for unmet care needs reported by at least 20% of the respondents.

All statistical procedures were performed using SPSS 24.0.

## Results

### Characteristics of the sample

Table 1 shows demographic characteristics of our two samples. Patients in the outpatient sample had significantly higher GAF-scores than those in the youth-ACT sample (mean = 54.7, sd = 5.5 vs. mean = 46.5, sd = 8.3). There were no significant differences between the outpatient sample and youth-ACT sample regarding age, gender, country of birth, type of ADHD diagnosis, and living situation.

The mean age in the outpatient sample (n = 52) was 11.2 years (sd = 2.8). A majority of the outpatients were boys (65.4%), most of whom were being raised in a two-parent household (69.2%). About two-thirds of outpatients (69.2%) had been diagnosed with ADHD combined type.

In the youth-ACT sample (n = 53), the patients' mean age was 12.3 years (sd = 3.2). As in the outpatient sample, a majority of youth-ACT sample consisted of boys (69.8%); and most patients (64.2%) were being raised in a two-parent household. A majority of patients in this subsample of youth-ACT patients had also been diagnosed with ADHD combined type (79.2%).

### Domains of needs

Using the ICF domains, the results of this study will be first described for the overall sample, followed by the results for the two different treatment settings separately. For reasons of brevity, we only highlight unmet care needs with a frequency of 15% or more in the text of this manuscript.

**Physical and mental functions.**   As Table 2 shows, mental health problems were the most frequently reported unmet care need in children and adolescents with ADHD: 61% reported an unmet need in this area (outpatient sample 66.0%; youth-ACT sample 57.7%; n.s.). The second most frequently reported unmet care need—which was reported by 47.6% of all patients—concerned information on diagnosis and treatment. 60.4% of the outpatient sample vs. 34.6%

**Table 1. Sample characteristics of the child or adolescent who received treatment.**

| | Outpatient | | Youth-ACT | | t-test/ corrected χ² -test (2-sided)/ Fisher's Exact test | P |
|---|---|---|---|---|---|---|
| | **Child (n = 52)** | | **Child (n = 53)** | | | |
| Age (sd) | Total mean | 11.2 (2.8) | Total mean | 12.3 (3.2) | -1,76$^t$ | 0.08 |
| | range | 6–17 | range | 6–17 | | |
| | Girls mean | 11.1 (3.4) | Girls mean | 13.8 (1.7) | | |
| | range | 6–17 | range | 10–17 | | |
| | Boys mean | 11.3 (2.5) | Boys mean | 11.6 (3.5) | | |
| | range | 6–15 | range | 6–17 | | |
| Gender | Girls | 34.6% | Girls | 30.2% | 0.08$^{χ2}$ | 0.782 |
| | Boys | 65.4% | Boys | 69.8% | | |
| Country of birth | The Netherlands | 100.0% | The Netherlands | 94.3% | | 0.243$^{FE}$ |
| | Other | 0.0% | Other | 5.7% | | |
| ADHD diagnosis | Combined | 69.2% | Combined | 79.2% | 0.90$^{χ2}$ | 0.342 |
| | Inattention | 30.8% | Inattention | 20.8% | | |
| GAF-score (sd) | Mean | 54.7 (5.5) | Mean | 46.5 (8.3) | 5.91$^t$ | 0.000 |
| | Range | 45–75 | Range | 15–55 | | |
| Living situation | Single parent | 30.8% | Single parent | 35.8% | 0.12 $^{χ2}$ | 0.730 |
| | Two parents | 69.2% | Two parents | 64.2% | | |

n = number of included patients

sd = standard deviation

GAF = General assessment of functioning

p = p-value; a value below 0.05 is considered to be statistically significant.

Independent sample t-test was performed to compare the mean score between the outpatient and youth-ACT samples with respect to continuous variable.

The χ² –test with a continuity correction was used to test the difference between the outpatient and youth-ACT sample with regard to a categorical variable with df = 1.

The Fisher's Exact test was performed because the number in at least one of the cells in the child or care provider sample was <5

doi: 10.6084/m9.figshare.11417070

of the youth-ACT sample reported this need (p < .05). Nearly a fifth of all ADHD patients (18.1%) perceived unmet care needs regarding medication-related side effects. This item differed significantly (p < .05) between the outpatient sample (9.4%) and youth-ACT sample (26.9%). Almost nine percent of all ADHD patients perceived unmet needs regarding the quality and/or quantity of food. Outpatients reported significantly fewer unmet care needs on this item than patients treated with youth-ACT (1.9% vs. 15.4%; p < .05).

**Performance of daily activities.** About 17% of all ADHD patients reported unmet care needs with respect to reading and writing skills. No significant differences were found between ACT-patients and regular outpatients. About ten percent of all ADHD-patients (10.5%) reported unmet needs pertaining to handling money, with no significant difference between the two samples. In the overall sample, about nine percent (8.6%) of the patients reported unmet care needs regarding their abilities for self-care (e.g., oral health, daily hygiene, and clothing). Patients in the youth-ACT sample reported significantly more unmet care needs on this item (15.4% vs. 1.9%; p < .05).

**Participation in the community.** Almost 29% of all ADHD patients in the overall sample perceived their future prospects (i.e., their opportunities/chances for a successful and prosperous life) as an unmet care need. Those referred for ACT-treatment reported unmet care needs in this area more frequently than those who we referred for regular treatment (38.5% vs. 18.9% respectively, p < .05).

**Table 2. Unmet Needs overview.**

| Unmet needs domains | Overall n (= 105) | % | Youth-ACT n (= 53) | % | Outpatient n (= 52) | % | Corrected $\chi^2$-test (2-sided)/ Fisher's Exact test | P |
|---|---|---|---|---|---|---|---|---|
| **Physical and mental functions** | | | | | | | | |
| Mental health problems (not psychotic) | 65 | 61.9 | 30 | 57.7 | 35 | 66.0 | 0.46 (df = 1) | 0.497 |
| Danger to others | 10 | 9.5 | 5 | 9.6 | 5 | 9.4 | 0.00 (df = 1) | 1.000 |
| Danger to themselves | 7 | 6.7 | 4 | 7.7 | 3 | 5.7 | FE | 0.716 |
| Psychotic symptoms | 7 | 6.7 | 2 | 3.8 | 5 | 9.4 | FE | 0.449 |
| Information regarding diagnosis/treatment | 50 | 47.6 | 18 | 34.6 | 32 | 60.4 | 5.99 (df = 1) | 0.014 |
| Physical handicap or disease | 6 | 5.7 | 3 | 5.8 | 3 | 5.7 | FE | 1.000 |
| Medication side effects | 19 | 18.1 | 14 | 26.9 | 5 | 9.4 | 4.30 (df = 1) | 0.038 |
| Drugs misuse/alcohol abuse | - | - | - | - | - | - | - | - |
| Food (qualitative or quantitative) | 9 | 8.6 | 8 | 15.4 | 1 | 1.9 | FE | 0.016 |
| **Performance of daily activities** | | | | | | | | |
| Reading/writing skills at expected grade level | 18 | 17.1 | 5 | 9.6 | 13 | 24.5 | 3.13 (df = 1) | 0.077 |
| Handling money | 11 | 10.5 | 5 | 9.6 | 6 | 11.3 | 0.00 (df = 1) | 1.000 |
| Self-care abilities (age-related) | 9 | 8.6 | 8 | 15.4 | 1 | 1.9 | FE | 0.016 |
| Paid job (including side jobs) | 7 | 6.7 | 4 | 7.7 | 3 | 5.7 | FE | 0.716 |
| Cleaning up room (or bedroom) | 5 | 4.8 | 2 | 3.8 | 3 | 5.7 | FE | 1.000 |
| Caring for someone else (family member or pet) | 1 | 1.0 | 1 | 1.9 | - | - | FE | 0.495 |
| **Participation in the community** | | | | | | | | |
| Regular/suitable school or other daytime activities | 21 | 20.0 | 16 | 30.8 | 5 | 9.4 | 6.19 (df = 1) | 0.013 |
| Making and/or keeping friends | 23 | 21.9 | 16 | 30.8 | 7 | 13.2 | 3.76 (df = 1) | 0.049 |
| Future prospects (opportunities/chances for a successful and prosperous life) | 30 | 28.6 | 20 | 38.5 | 10 | 18.9 | 4.02 (df = 1) | 0.045 |
| Access to (public) transport | 5 | 4.8 | 3 | 5.8 | 2 | 3.8 | FE | 0.678 |
| Housing | 3 | 2.9 | 2 | 3.8 | 1 | 1.9 | FE | 0.618 |
| Access to modern tools of communication | 2 | 1.9 | 1 | 1.9 | 1 | 1.9 | FE | 1.000 |
| Intimate relations | 4 | 3.8 | 3 | 5.8 | 1 | 1.9 | FE | 0.363 |
| Sexuality | 1 | 1.0 | 1 | 1.9 | - | - | FE | 0.495 |

n = number of included patients

p = p-value; a value below 0.05 is considered to be statistically significant. The χ2 –test with a continuity correction was performed because df = 1. Fisher's Exact test was performed if the number in at least one of the cells in the youth-ACT or outpatient sample was <5

FE = Fisher's Exact test

doi: 10.6084/m9.figshare.11417070

More than a fifth of the ADHD patients in the overall sample (21.9%) perceived unmet needs regarding making and/or keeping friends, with a significant difference between the youth-ACT sample (30.8%) and the outpatient sample (13.2%; p < .05). Twenty percent of all ADHD patients reported unmet needs with respect to having regular and suitable school or other daytime activities (e.g., practicing a sport/hobby). The scores between the outpatient (9.4%) and youth-ACT samples (30.8%) differed significantly (p < .05).

## Comparing children and adolescents

In the overall sample, no significant differences were found between primary school children (age 6–12 years) and adolescents (age 13–17 years) regarding the five most frequently reported unmet care needs: mental health problems, information on diagnosis and treatment, having regular and suitable school or other daytime activities, making and/or keeping friends, and future prospects.

## Discussion

Non-compliance with, or drop-out from treatment, result in suboptimal results of treatment of childhood ADHD. We therefore studied unmet care needs in children and adolescents with ADHD in two treatment settings. Compared to young adults, in whom unmet care needs in various areas were found to a frequency of up to thirty percent [27], children and adolescents with ADHD reported levels of unmet needs up to sixty percent. Further, differences in frequencies of unmet needs were found between an outpatient versus a youth-ACT sample. To our knowledge, this is the first study regarding unmet care needs in children and adolescents with ADHD in such samples.

### Overall sample

We found considerable variations in the frequencies with which ADHD patients reported unmet care needs. The three self-reported unmet care needs most reported by children and adolescents with ADHD lay in two domains: (1) mental health problems and information on diagnosis and/or treatment (in the domain of physical and mental functions); and (2) future prospects (in the domain of participation in the community).

   Given the admission of these patients to a specialized mental health treatment setting, the high number reporting these unmet care needs is what one would expect. For the same reason, however, it is striking that so many of the children and adolescents with ADHD reported no unmet care needs related to mental health problems (40%), information on diagnosis and/or treatment (50%), and future prospects (70%). Various explanations for this are possible. For example, these patients—who had already been referred to specialized mental healthcare—were aware of their problems, but considered the help they were receiving to be sufficient (meaning that their needs were being met). Alternatively, unlike their parents or mental healthcare providers, these patients had been unaware of their problems, which is why others had had to take the initiative for their treatment.

   In clinical practice, such potential discrepancies in perception are significant. Our findings indicate that if a mental health problems is objectively diagnosed by a clinician, it is simultaneously important to determine whether there are differences in perceptions on the patient's (mental) health problems [43, 44]. If present, such differences may reduce the quality of any agreement between patient and mental health professionals on treatment goals and treatment options (tasks) during treatment [45]. Clarifying any possible different perceptions of care needs and exploring differing perceptions of necessary treatment may help prevent non-attendance, non-compliance and drop-out [46, 47].

   No significant differences were found between primary school children and adolescents regarding the five most frequently perceived unmet care needs: mental health problems, information on diagnosis and treatment, having regular and suitable school or other daytime activities, making and/or keeping friends, and future prospects. This is remarkable because we assumed that adolescents, due to their cognitive development and decrease of parental support, would be more aware of their problems and therefore would perceive more unmet care needs than younger children [48]. Further, the nature of unmet care needs might change across development. The lack of significant age effects may indicate that young children may be as aware of unmet care needs as adolescents. Further, in their desire for autonomy [48], adolescents may under-report unmet care needs.

   Another interesting finding is that ADHD patients perceived no unmet care needs for drug misuse or alcohol abuse. This is remarkable because ADHD often co-occurs with substance abuse and dependence (e.g. cannabis misuse) [49–51]. Several factors may explain why children and adolescents expressed no unmet care needs in this area. It may be that actual use was

relatively low in our sample because patients with problematic drug misuse or alcohol abuse were referred to specialized drug treatment centres. But it is also possible that patients with problematic alcohol or substance abuse did not perceive their use as a problem.

## Comparing outpatient clinics with youth-ACT

Our comparison of outpatient clinics and youth-ACT revealed significant differences between settings regarding a quarter of the unmet care needs we investigated. In line with our a priori hypotheses, ADHD patients from the youth-ACT setting reported significantly more unmet care needs than those treated in the general outpatient care setting. The notable exception, in the domain of physical and mental functions, was that outpatients with ADHD were more likely than those in the youth-ACT sample to perceive unmet needs with respect to information on diagnosis and treatment.

The differences between the two settings regarding unmet needs could not be explained by age, gender, type of ADHD diagnosis, living situation or country of birth. However, comparison between the two treatment settings showed a significant difference regarding the GAF-score, indicating that ACT patients had more problems in daily functioning [2].

For the purpose of conciseness, only the results with the most clinical relevance will be highlighted now.

**More frequent unmet needs in outpatient clinics.** Outpatients with ADHD were more likely than those in the youth-ACT sample to perceive unmet needs with respect to information on treatment. One possible explanation for this is that patients in the youth-ACT setting had already received this information during their previous outpatient treatment, whereas many outpatients who had recently started treatment had not. Another possible explanation is that ADHD patients in the youth-ACT setting were less interested in obtaining information on treatment because of limited engagement in treatment.

As patients' treatment adherence can be significantly improved by obtaining relevant information on treatment options and possible outcomes, we recommend that care providers investigate whether patients need such information. We also recommend that care providers investigate why a patient does not report a need for information [47]. Treatment adherence and treatment outcome may be improved by a process of shared decision-making based on shared information [52, 53].

For clinical practice, our findings suggest that many patients consider themselves uninformed about assigned diagnoses (60%) in the general outpatient group, and one third in the youth-ACT group. In both settings, clinicians should pay close attention to providing in information about diagnosis and treatment.

**More frequent unmet needs in Youth-ACT.** In the domain of physical and mental functions, side effects of medication were perceived significantly more by youth-ACT patients than by outpatients. Given the severity of their psychiatric problems, it may be that ADHD patients in the youth-ACT setting are more likely to perceive side effects, because their treatment requires more intensive medication. It is likely that the side effects of medication they experience have a negative impact on medication compliance, and, in turn, on treatment outcome [54]. A particular recommendation for professionals in youth-ACT settings is to thoroughly identify such side effects. If necessary, action can be taken to reduce them.

With further regard to the domain of physical and mental functions, significantly more patients with ADHD in the youth-ACT setting perceived unmet needs with respect to food quality and quantity. Unmet needs in this area were reported by 15.4% of the youth-ACT sample. A possible explanation is that children treated with ACT often grow up in families with limited financial resources and more financial problems, which can lead to less healthy food

patterns [11–13]. Because more than one out of ten youth-ACT patients with ADHD reported problems with food, we recommend that clinicians who treat the most vulnerable ADHD patients, in youth-ACT samples or other high-risk samples such as inpatient samples, routinely assess needs in this area. Lack of healthy food attenuates psychological and social functioning, and may influence motivation for treatment, which in turn could lead to suboptimal treatment outcome [55].

We should also draw attention to the high level of unmet care needs related to participation in the community in the youth-ACT sample. This finding is in line with our a priori hypotheses. The largest difference between patients from the two settings involved participation in the community. Recipients of youth-ACT perceived more unmet care needs in this area. As youth-ACT focuses specifically on enhancing patients' societal functioning, this score indicates that most of these patients had been referred to the appropriate treatment setting.

A high number of youth-ACT patients in this study reported unmet needs with regard to future prospects, regular and/or suitable school or other daytime activities, and making and/or keeping friends. Problems in these areas may potentially threaten a young person's development. Hence, it is important that healthcare providers, especially those in youth-ACT settings, identify the causes underlying these problems, and subsequently initiate treatment interventions that are likely to meet the unmet care needs in question [48, 55–58]. For children and adolescents with ADHD belonging to a high-risk sample, such as those who are treated with youth-ACT, this implicates that routine assessment of school functioning, being one of the hallmarks of state-of-the-art investigation, may not be enough. Broader assessment of societal functioning, including patients' views on chances in society (future prospects), daytime activities, and abilities to make or keep friends, may be needed if regular outpatient treatment is not successful. Because patients report high frequencies of unmet needs in these areas, targeting these factors may ameliorate treatment outcome. In other words, in high-risk ADHD patients, drug treatment and other—merely—symptom focused interventions may not be sufficient.

## Strengths and limitations

This is the first study to investigate the self-reported perceived care needs of children and adolescents with ADHD who had been referred to general outpatient care or youth-ACT. Our inclusion of the latter in our sample enabled us to examine the perceived unmet care needs of ADHD patients with severe psychiatric and psychosocial problems who, after failing to respond to regular interventions, had been referred to more intensive youth-ACT treatment. By comparing the perceived unmet care needs between the two samples, we were thus able to provide insight into the unmet needs of children with ADHD receiving treatment in two treatment settings characterized by different intensities of treatment. This enabled us to study the influence of severity of psychiatric and psychosocial problems on unmet care needs.

A clear limitation is the cross-sectional design of the study, which prevented us from providing causal explanations for the occurrence and persistence of unmet care needs.

## Conclusions

In summary, the three most important unmet care needs perceived by ADHD patients concerned mental health problems, information on diagnosis and/or treatment, and future prospects. While outpatients perceived more unmet care needs regarding information on diagnosis/treatment, those treated within the youth-ACT setting reported more unmet needs concerning medication side effects, quality and/or quantity of food, self-care abilities, regular/ suitable school or other daytime activities, making and/or keeping friends, and future prospects. Our data suggest that focusing treatment of ADHD patients on unmet needs, and not

only on ADHD symptoms, may motivate patients, and may reduce non-attendance, non-compliance, and drop-out. It remains to be tested whether a needs-led approach would indeed improve treatment outcome.

## Acknowledgments

The authors gratefully acknowledge the families and mental healthcare providers who participated, and also acknowledge research assistant Amanda Noorman, who helped conduct the study. We thank Nannet Buitelaar, Adriaan Hoogendoorn, and David Alexander for their valuable comments on this paper. We would also like to thank Daphne van de Draai for her assistance with building the SPSS-files.

## Author Contributions

**Conceptualization:** Richard Vijverberg.

**Formal analysis:** Richard Vijverberg.

**Investigation:** Richard Vijverberg.

**Methodology:** Richard Vijverberg, Robert Ferdinand, Aartjan Beekman, Berno van Meijel.

**Project administration:** Richard Vijverberg.

**Supervision:** Robert Ferdinand, Aartjan Beekman, Berno van Meijel.

**Writing – original draft:** Richard Vijverberg, Robert Ferdinand.

**Writing – review & editing:** Aartjan Beekman, Berno van Meijel.

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
