## [Decision Letter · Decision Letter 0]

15 Oct 2019

PONE-D-19-19638

Unmet Care Needs of Children with ADHD: A Cross-sectional Study Comparing a General Outpatient Setting with Youth Assertive Community Treatment.

PLOS ONE

Dear Dr Vijverberg,

Thank you for submitting your manuscript to PLOS ONE. After careful consideration, we feel that it has merit but does not fully meet PLOS ONE’s publication criteria as it currently stands. Therefore, we invite you to submit a revised version of the manuscript that addresses the points raised during the review process.

We would appreciate receiving your revised manuscript by the 15th December 2019. To enhance the reproducibility of your results, we recommend that if applicable you deposit your laboratory protocols in protocols.io, where a protocol can be assigned its own identifier (DOI) such that it can be cited independently in the future. For instructions see: http://journals.plos.org/plosone/s/submission-guidelines#loc-laboratory-protocols

We look forward to receiving your revised manuscript.

Kind regards,

Michelle Tye, Ph.D.

Academic Editor

PLOS ONE

Journal Requirements:

Additional Editor Comments:

No additional comments. 

Reviewers' comments:

Reviewer's Responses to Questions

**Comments to the Author**

1. Is the manuscript technically sound, and do the data support the conclusions?

Reviewer #1: Yes

Reviewer #2: Yes

2. Has the statistical analysis been performed appropriately and rigorously? 

Reviewer #1: Yes

Reviewer #2: Yes

3. Have the authors made all data underlying the findings in their manuscript fully available?

Reviewer #1: Yes

Reviewer #2: No

4. Is the manuscript presented in an intelligible fashion and written in standard English?

Reviewer #1: Yes

Reviewer #2: Yes

5. Review Comments to the Author

Reviewer #1: This is a very well-written manuscript, current and relevant background information. Very interesting results on the unmet care needs for children in outpatient setting compared to the ones on the youth-ACT.

I would like to see more information on the differences in perceptions of unmet care needs by age, since at younger ages, parents are more involved on their children's care and treatment compared to adolescents.

Reviewer #2: An interesting study where, as I perceive it, the aim was to study unmet needs to a wide extent, not only medical needs but also ICF aspects: (a) physical and mental functions, (b) performance of daily activities, (c) participation in the community, in children and adolescents with ADHD.

To this end, two groups were recruited, one with contact with specialized Child and Adolescent Psychiatric (CAP) care (who presumably should have good help with their needs) and a psycho-socially vulnerable group that received intensive and outreach-oriented mental health care for patients with more severe psychiatric and psychosocial problems (ATC-group) where you could expect that the participants had more unmet needs due to e.g. limited financial resources and more financial problems.

This ambition of the study is excellent and the findings are well worth to be published. The study has reasonably large groups and established measurement instruments were used. In the background section the state of knowledge in the area (which is limited) is well described, there are good arguments for the importance of identifying unmet needs, and the point of discovering such needs is obviously important for both patients’ well-being and also, as the authors point out, are likely to affect adherence to treatment.

Still I have a number of comments:

1/ A general comment is that it is not quite clear to me what the main focus of the study is; is it to direct attention to unmet needs in children with ADHD in general, or to investigate potential differences between patient groups depending on their general psychosocial situation, where the ATC-group is presumed to be less well-off. I think that the authors in the Results and Discussion sections give the impression that they want to cover both aspects, something which I support, but that is not congruent with the title “Unmet Care Needs of Children with ADHD: A Cross-sectional Study Comparing a General Outpatient Setting with Youth Assertive Community Treatment.” I think the Short title: ”Unmet Care Needs of Children with ADHD” would be better. In that case the unexplained mentioning of the method ”Youth Assertive Community Treatment” would also not confuse the reader.

2/ If the authors agree that both unmet needs in ADHD in general, and differences with regard to general psychosocial situation are in focus, then I would suggest that reporting of results and the disposition of the discussion section are organized into the areas where ACT-group is worse and where CAP is worse (and those where they are equal).

3/ Abstract:

I think the abstract lacks important information in order for the reader to grasp the study. Ages of participants, number of participants, etc. According to author guidelines (checked 2019-10-11 of PLOS ONE) there is room for another 175 words.

What is Youth Assertive Community Treatment? Shouldn't that be explained in the abstract? And that the ACT group is thought to be psychosocially burdened?

In Results it is stated that “Compared to youth-ACT patients, outpatients perceived more unmet needs regarding daily activities”. As I can see in table 2 the only item where CAP outpatients scored higher was “Reading/writing skills at expected grade level” p=0.077, while ATC-group scored higher on ”Self-care abilities (age-related)” p=0.016. Please reconsider what information about findings you would like to put forward in the abstract.

In Conclusions it is stated that “Focusing treatment of ADHD patients on unmet needs that were detected may reduce…”. Perhaps it is a mistake in wording, but it is not said in what way the clinician should detect the unmet needs.

4/ As I mentioned above the Background section is well written. However reference number 1 is a bit old. This one is more updated: Polanczyk, G. V., Salum, G. A., Sugaya, L. S., Caye, A., & Rohde, L. A. (2015). Annual research review: A meta-analysis of the worldwide prevalence of mental disorders in children and adolescents. J Child Psychol Psychiatry, 56(3), 345-365. doi:10.1111/jcpp.12381.

The authors argue at the bottom of page 9 that the perspective of patients with ADHD is not much investigated. However, there is at least one study: Emilsson et al. Beliefs regarding medication and side effects influence treatment adherence in adolescents with attention deficit hyperactivity disorder. Eur Child Adolesc Psychiatry 2017;26:559-571.

5/ On Page 10:

Design

“This cross-sectional naturalistic study was conducted between 2015 and 2017 in a specialized treatment center for child and adolescent psychiatric disorders in the Netherlands.”

– but there were 2 different settings! It is presented in the next paragraph, but it becomes contradictory.

“Setting

Participants were recruited from two general outpatient treatment settings and one youth-ACT team.”

- And are the two general outpatient treatment settings part of the specialized treatment center? On what organisational level are the general outpatient treatment settings? Are they at a specialized level so that you need to be referred there? Which means that you already have had a previous medical contact.

I also think it would be appropriate to describe the presumption that the ACT-group was less psycho-socially well-off here.

6/ Something which is not mentioned at all is cannabis consumption, something that could aggravate ADHD symptoms and treatment. Is there anything known about that? More cannabis use in the ACT-group?

7/ Page 11

Participants

“…. A random sample was selected from the general outpatient population.”

- Was the investigation of unmet needs done directly when they came on their first visit, or did they have ongoing contact? - If so, for how long / how many visits?

“For the youth-ACT sample, we included all patients who were referred to this treatment setting during the inclusion period. These ACT-patients all had received prior general outpatient treatment.”

- Does this mean that they had had contact with general CAP outpatient treatment? Or with primary care?

8/ It is good that a flow chart is provided. However, I would suggest that the total number of patients that the randomisation was based on in the general outpatient population also was presented, i.e. a re-writing of Figure 1.

9/ Page 12 Measurement instruments

- Was MINI-KID made as part of the research investigation or was it routine measure for clinical intake? Who did the MINI-KID? Did you have any supplementary diagnostic measures like questionnaires from parents, teachers, etc?

- The CANSAS is said to have been administered in a face-to-face interview with the patient during the intake procedure. – Were 6-years old interviewed? Did the parents participate? Is CANSAS validated for children?

10/ - When were the interviews done? It is reported that they were done in “…outpatients who had recently started treatment”, but it does matter how long they have been in treatment, doesn’t it? If the patients are still uninformed about their diagnosis after having received a certain amount of treatment is quite different from if it was at their first appointment.

11/ Page 13. There are surprisingly small differences (except for GAF) between the two groups as shown in Table 1. Could you comment on that, please? The presumption is that the ACT-group had more severe psychiatric and psychosocial problems, but that can’t be concluded from table 1.

12/ Page 14

Results

I think that there is an unnecessary careful review of findings of unmet needs, is it not enough with details in table 2? And you could instead in the text highlight interesting findings and the differences between the groups.

Physical and mental functions

As I wrote above I would suggest that in reporting of results the text would improve by being organized in the areas where ACT-group is worse and where CAP is worse (and those where they are equal).

The fact that patients who have come to specialized CAP care to such an extent (60%) considered themselves uninformed about diagnosis is remarkable, a major finding that should affect care routines and this should be mentioned as a first finding and further emphasized in the discussion.

Performance of daily activities

In this section the presentation works well as it stands but in:

Page 15 Participation in the community, I suggest the same organisation of findings as for Physical and mental functions above.

13/ Page 18 Discussion

”… Overall, ADHD patients reported substantial levels of unmet needs in several areas…”

The authors argue that they found substantial levels of unmet needs. But how many percent indicates high or low? Are there any studies to compare with?

Emphasize that patients who have come to specialized CAP care to such an extent (60%) considered themselves uninformed about diagnosis.

The ACT-group is almost always disadvantaged in the findings, probably as a function of being disadvantaged in several areas. Highlight that in the beginning of the introduction section.

You indicate on page 10 that the study was driven by two a priori hypotheses. Please comment on if your findings support those hypotheses.

6. PLOS authors have the option to publish the peer review history of their article (what does this mean?). If published, this will include your full peer review and any attached files.

Reviewer #1: No

Reviewer #2: Yes: Per A Gustafsson

---

## [Author Response · Author response to Decision Letter 0]

20 Dec 2019

Feedback reviewer 1

Comment 1: 

This is a very well-written manuscript, current and relevant background information. Very interesting results on the unmet care needs for children in outpatient setting compared to the ones on the youth-ACT.

Response 1: 

Thank you.

Comment 2: 

I would like to see more information on the differences in perceptions of unmet care needs by age, since 

at younger ages, parents are more involved on their children's care and treatment compared to 

adolescents.

Response 2: 

We agree with the reviewer that perceptions of unmet care needs may depend on age. In accordance with the suggestions of the reviewer, we have chosen to distinguish between (primary school) children (age 6-12 years) and adolescents (age 13-17 years). For the total sample, we performed the Chi-square test with Yates continuity correction (= χ2 -test) to analyse the differences between primary school children and adolescents regarding unmet care needs with a frequency of 20% or more (see Table 3). In the main text we added the following information:

We added (background section, page 4.):

“This comparison enabled us to judge the influence of severity of psychiatric and psychosocial problems on unmet care needs [13]. Further, since parents are involved differently in younger versus older children, we investigated unmet care needs in two age groups: primary school children and adolescents.”

In the method section we added the following text (method section. data-analysis, page 7):

“To investigate the association between age and unmet care needs, we constructed two subgroups, i.e., primary school children (age 6-12 years) and adolescents (age 13-17 years). For the overall sample, we performed the Chi-square test with Yates continuity correction (= χ2 -test) to analyse differences between these two age groups for unmet care needs reported by at least 20% of the respondents.”

We added (result section, page 13.):

“Comparing children and adolescents

In the overall sample, no significant differences were found between primary school children (age 6-12 years) and adolescents (age 13-17 years) regarding the five most frequently reported unmet care needs: mental health problems, information on diagnosis and treatment, having regular and suitable school or other daytime activities, making and/or keeping friends, and future prospects.”

We added (discussion section, page 14.):

“No significant differences were found between primary school children and adolescents regarding the five most frequently perceived unmet care needs: mental health problems, information on diagnosis and treatment, having regular and suitable school or other daytime activities, making and/or keeping friends, and future prospects. This is remarkable because we assumed that adolescents, due to their cognitive development and decrease of parental support, would be more aware of their problems and therefore would perceive more unmet care needs than younger children [48]. Further, the nature of unmet care needs might change across development. The lack of significant age effects may indicate that young children may be as aware of unmet care needs as adolescents. Further, in their desire for autonomy [48], adolescents may under-report unmet care needs.”

Feedback reviewer 2

Comment 1: 

An interesting study where, as I perceive it, the aim was to study unmet needs to a wide extent, not only medical needs but also ICF aspects: (i) physical and mental functions, (ii) performance of daily activities, (iii) participation in the community, in children and adolescents with ADHD. To this end, two groups were recruited, one with contact with specialized Child and Adolescent Psychiatric (CAP) care (who presumably should have good help with their needs) and a psycho-socially vulnerable group that received intensive and outreach-oriented mental health care for patients with more severe psychiatric and psychosocial problems (ATC-group) where you could expect that the participants had more unmet needs due to e.g. limited financial resources and more financial problems.

This ambition of the study is excellent and the findings are well worth to be published. The study has reasonably large groups and established measurement instruments were used. In the background section the state of knowledge in the area (which is limited) is well described, there are good arguments for the importance of identifying unmet needs, and the point of discovering such needs is obviously important for both patients’ well-being and also, as the authors point out, are likely to affect adherence to treatment.

Response 1: 

First, we warmly thank the reviewer for his/her very useful comments and suggestions. 

Comment 2: 

A general comment is that it is not quite clear to me what the main focus of the study is; is it to direct attention to unmet needs in children with ADHD in general, or to investigate potential differences between patient groups depending on their general psychosocial situation, where the ATC-group is presumed to be less well-off. I think that the authors in the Results and Discussion sections give the impression that they want to cover both aspects, something which I support, but that is not congruent with the title “Unmet Care Needs of Children with ADHD: A Cross-sectional Study Comparing a General Outpatient Setting with Youth Assertive Community Treatment.” I think the Short title: Unmet Care Needs of Children with ADHD would be better. In that case the unexplained mentioning of the method ”Youth Assertive Community Treatment” would also not confuse the reader.

Response 2: 

This study primarily investigated unmet care needs in ADHD patients in general. A second aim was to 

distinguish between two treatment settings with different treatment intensities, enabling us to assess 

the influence of severity of psychiatric and psychosocial problems. The study covers both aspects, and 

therefore, we changed the title according to the suggestion of the reviewer: 

Original text (title, page 1.): 

“Unmet needs of children with ADHD: a cross-sectional study comparing a general outpatient 

setting with youth assertive community treatment”

New text:

“Unmet care needs of children with ADHD”

Further, we changed the abstract.

Original text (abstract, page 2.):

“Background: Non-attendance, non-compliance, or drop-out from treatment, result in suboptimal treatment of childhood ADHD. These problems may be due to mismatches between needs of patients and treatments provided.”

New text:

“Background: Non-compliance to, or drop-out from treatment for childhood ADHD, result in suboptimal outcome. Non-compliance and drop-out may be due to mismatches between patients' care needs and treatments provided. This study investigated unmet care needs in ADHD patients. Unmet needs were assessed in two different treatment settings (general outpatient setting versus youth-ACT). Youth-ACT treatment is an intensive outreach-oriented treatment for patients with severe psychiatric and psychosocial problems. Comparison of a general outpatient sample with a youth-ACT sample enabled us to assess the influence of severity of psychiatric and psychosocial problems on perceived care needs.”

Further, to emphasize the focus of the study we added the following sentence.

We added (discussion section, strengths and limitations, page 17.):

“Our inclusion of the latter in our sample enabled us to examine the perceived unmet care needs of ADHD patients with severe psychiatric and psychosocial problems who, after failing to respond to regular interventions, had been referred to more intensive youth-ACT treatment.”

Comment 3 (results section):

I think that there is an unnecessary careful review of findings of unmet needs, is it not enough with details in table 2? And you could instead in the text highlight interesting findings and the differences between the groups. `Physical and mental functions’; I would suggest that in reporting of results the text would improve by being organized in the areas where ACT-group is worse and where CAP is worse (and those where they are equal).‘Performance of daily activities’, in this section the presentation works well as it stands. ‘Participation in the community`: I suggest the same organization of findings as for Physical and mental functions above.

Response 3:

We agree with the reviewer that we provided a very detailed description of our findings. In accordance with the suggestion of the reviewer, we now only highlight the most interesting findings. We now only report unmet care needs with a frequency of 15% or more. In line with the reviewer's suggestion, for each ICF domain we first report the results for the total sample (from high to low frequency), and next the most frequently reported unmet care needs for each subgroup separately.

New text (result section, domains of needs, page 9-10.):

“Domains of needs

Using the ICF domains, the results of this study will be first described for the overall sample, followed by the results for the two different treatment settings separately. For reasons of brevity, we only highlight unmet care needs with a frequency of 15% or more in the text of this manuscript.

Physical and mental functions 

As Table 2 shows, mental health problems were the most frequently reported unmet care need in children and adolescents with ADHD: 61% reported an unmet need in this area (outpatient sample 66.0%; youth-ACT sample 57.7%; n.s.). The second most frequently reported unmet care need—which was reported by 47.6% of all patients—concerned information on diagnosis and treatment. 60.4% of the outpatient sample vs. 34.6% of the youth-ACT sample reported this need (p < .05). Nearly a fifth of all ADHD patients (18.1%) perceived unmet care needs regarding medication-related side effects. This item differed significantly (p < .05) between the outpatient sample (9.4%) and youth-ACT sample (26.9%). Almost nine percent of all ADHD patients perceived unmet needs regarding the quality and/or quantity of food. Outpatients reported significantly fewer unmet care needs on this item than patients treated with youth-ACT (1.9% vs. 15.4%; p < .05). 

Performance of daily activities

About 17% of all ADHD patients reported unmet care needs with respect to reading and writing skills. No significant differences were found between ACT-patients and regular outpatients. About ten percent of all ADHD-patients (10.5%) reported unmet needs pertaining to handling money, with no significant difference between the two samples. In the overall sample, about nine percent (8.6%) of the patients reported unmet care needs regarding their abilities for self-care (e.g., oral health, daily hygiene, and clothing). Patients in the youth-ACT sample reported significantly more unmet care needs on this item (15.4% vs. 1.9%; p < .05). 

Participation in the community

Almost 29% of all ADHD patients in the overall sample perceived their future prospects (i.e., their opportunities/chances for a successful and prosperous life) as an unmet care need. Those referred for ACT-treatment reported unmet care needs in this area more frequently than those who we referred for regular treatment (38.5% vs. 18.9% respectively, p < .05).

More than a fifth of the ADHD patients in the overall sample (21.9%) perceived unmet needs regarding making and/or keeping friends, with a significant difference between the youth-ACT sample (30.8%) and the outpatient sample (13.2%; p < .05). Twenty percent of all ADHD patients reported unmet needs with respect to having regular and suitable school or other daytime activities (e.g., practicing a sport/hobby). The scores between the outpatient (9.4%) and youth-ACT samples (30.8%) differed significantly (p < .05).”

Comment 4: 

If the authors agree that both unmet needs in ADHD in general, and differences with regard to general 

psychosocial situation are in focus, then I would suggest that reporting of results and the disposition of the discussion section are organized into the areas where ACT-group is worse and where CAP is worse (and those where they are equal).

Response 4: 

In accordance with the suggestion of the reviewer, we re-organized the discussion section “comparing outpatient clinics with youth-ACT” and now describe the areas where ACT-group is worse, and then where outpatient-group is worse.

New text (discussion section, page 15.):

“Comparing outpatient clinics with youth-ACT.

Our comparison of outpatient clinics and youth-ACT revealed significant differences between settings regarding a quarter of the unmet care needs we investigated. In line with our a priori hypotheses, ADHD patients from the youth-ACT setting reported significantly more unmet care needs than those treated in the general outpatient care setting. The notable exception, in the domain of physical and mental functions, was that outpatients with ADHD were more likely than those in the youth-ACT sample to perceive unmet needs with respect to information on diagnosis and treatment.

The differences between the two settings regarding unmet needs cloud not be explained by age, gender, type of ADHD diagnosis, living situation or country of birth. However, comparison between the two treatment settings showed a significant difference regarding the GAF-score, indicating that ACT patients had more problems in daily functioning [2].

For the purpose of conciseness, only the results with the most clinical relevance will be highlighted now.

More frequent unmet needs in outpatient clinics

Outpatients with ADHD were more likely than those in the youth-ACT sample to perceive unmet needs with respect to information on treatment. One possible explanation for this is that patients in the youth-ACT setting had already received this information during their previous outpatient treatment, whereas many outpatients who had recently started treatment had not. Another possible explanation is that ADHD patients in the youth-ACT setting were less interested in obtaining information on treatment because of limited engagement in treatment. 

As patients’ treatment adherence can be significantly improved by obtaining relevant information on treatment options and possible outcomes, we recommend that care providers investigate whether patients need such information. We also recommend that care providers investigate why a patient does not report a need for information [47]. Treatment adherence and treatment outcome may be improved by a process of shared decision-making based on shared information [52, 53].

For clinical practice, our findings suggest that many patients consider themselves uninformed about assigned diagnoses (60%) in the general outpatient group, and one third in the youth-ACT group. In both settings, clinicians should pay close attention to providing in information about diagnosis and treatment.

More frequent unmet care needs in youth-ACT 

In the domain of physical and mental functions, side effects of medication were perceived significantly more by youth-ACT patients than by outpatients. Given the severity of their psychiatric problems, it may be that ADHD patients in the youth-ACT setting are more likely to perceive side effects, because their treatment requires more intensive medication. It is likely that the side effects of medication they experience have a negative impact on medication compliance, and, in turn, on treatment outcome [54]. A particular recommendation for professionals in youth-ACT settings is to thoroughly identify such side effects. If necessary, action can be taken to reduce them. 

With further regard to the domain of physical and mental functions, significantly more patients with ADHD in the youth-ACT setting perceived unmet needs with respect to food quality and quantity. Unmet needs in this area were reported by 15.4% of the youth-ACT sample. A possible explanation is that children treated with ACT often grow up in families with limited financial resources and more financial problems, which can lead to less healthy food patterns [11-13]. Because more than one out of ten youth-ACT patients with ADHD reported problems with food, we recommend that clinicians who treat the most vulnerable ADHD patients, in youth-ACT samples or other high-risk samples such as inpatient samples, routinely assess needs in this area. Lack of healthy food attenuates psychological and social functioning, and may influence motivation for treatment, which in turn could lead to suboptimal treatment outcome [55]. 

We should also draw attention to the high level of unmet care needs related to participation in the community in the youth-ACT sample. This finding is in line with our a priori hypotheses. The largest difference between patients from the two settings involved participation in the community. Recipients of youth-ACT perceived more unmet care needs in this area. As youth-ACT focuses specifically on enhancing patients' societal functioning, this score indicates that most of these patients had been referred to the appropriate treatment setting.

A high number of youth-ACT patients in this study reported unmet needs with regard to future prospects, regular and/or suitable school or other daytime activities, and making and/or keeping friends. Problems in these areas may potentially threaten a young person’s development. Hence, it is important that healthcare providers, especially those in youth-ACT settings, identify the causes underlying these problems, and subsequently initiate treatment interventions that are likely to meet the unmet care needs in question [48, 55-58]. For children and adolescents with ADHD belonging to a high-risk sample, such as those who are treated with youth-ACT, this implicates that routine assessment of school functioning, being one of the hallmarks of state-of-the-art investigation, may not be enough. Broader assessment of societal functioning, including patients’ views on chances in society (future prospects), daytime activities, and abilities to make or keep friends, may be needed if regular outpatient treatment is not successful. Because patients report high frequencies of unmet needs in these areas, targeting these factors may ameliorate treatment outcome. In other words, in high-risk ADHD patients, drug treatment and other—merely—symptom focused interventions may not be sufficient.”

Comment 5: 

It is good that a flow chart is provided. However, I would suggest that the total number of patients that the randomization was based on in the general outpatient population also was presented, i.e. a re-writing of Figure 1.

Response 5: 

We agree with the reviewer's comment that the total number of patients on whom randomization was 

based in the general outpatient population should be presented in the flow chart. Therefore, we re-wrote Figure 1: see revised manuscript.

Comment 6 (abstract, page 2):

In Results abstract it is stated that “Compared to youth-ACT patients, outpatients perceived more unmet needs regarding daily activities”. As I can see in table 2 the only item where CAP outpatients scored higher was “Reading/writing skills at expected grade level” p=0.077, while ATC-group scored higher on ”Self-care abilities (age-related)” p=0.016. Please reconsider what information about findings you would like to put forward in the abstract.

Response 6:

We agree that it was confusing and that is why we have improved the text. Indeed, there are hardly any differences with regard to daily activities in general. Therefore, those items where a significant difference was found are now discussed separately.

Original text (abstract, page 2): 

“Results: Unmet needs regarding mental health problems, information on diagnosis/treatment, and future prospects were reported most frequently. Compared to youth-ACT patients, outpatients perceived more unmet needs regarding daily activities. Youth-ACT patients reported more unmet needs concerning participation in the community.”

New text:

“Results: ADHD patients most frequently reported unmet needs regarding mental health problems, information on diagnosis/treatment, and future prospects. Outpatients differed from youth-ACT patients with respect to 30% of the unmet care needs that were investigated. Outpatients perceived more unmet needs regarding information on diagnosis/treatment (p=0.014). Youth-ACT patients perceived more unmet needs concerning medication side effects (p=0.038), quality and/or quantity of food (p=0.016), self-care abilities (p=0.016), regular/suitable school or other daytime activities (p=0.013), making and/or keeping friends (p=0.049), and future prospects (p=0.045).”

Original text (conclusions, page 19):

“In summary, the three main unmet care needs perceived by ADHD patients concerned mental health problems, information on diagnosis and/or treatment, and future prospects. While outpatients perceived more unmet care needs regarding daily activities, those treated within the youth-ACT setting reported more unmet needs concerning participation in the community. Our data suggest that focusing treatment of ADHD patients on the unmet needs may reduce non-attendance, non-compliance, and drop-out. It remains to be tested whether a needs-led approach would improve treatment outcomes.” 

New text:

“In summary, the three most important unmet care needs perceived by ADHD patients concerned mental health problems, information on diagnosis and/or treatment, and future prospects. While outpatients perceived more unmet care needs regarding information on diagnosis/treatment, those treated within the youth-ACT setting reported more unmet needs concerning medication side effects, quality and/or quantity of food, self-care abilities, regular/suitable school or other daytime activities, making and/or keeping friends, and future prospects. Our data suggest that focusing treatment of ADHD patients on unmet needs, and not only on ADHD symptoms, may motivate patients, and may reduce non-attendance, non-compliance, and drop-out. It remains to be tested whether a needs-led approach would indeed improve treatment outcome.”

Comment 7: 

Something which is not mentioned at all is cannabis consumption, something that could aggravate ADHD symptoms and treatment. Is there anything known about that? More cannabis use in the ACT-group?

Response 7: 

In the literature, cannabis use has been identified as a comorbid problem in patients with ADHD in the age group above 12 years [1, 2, 3]. In our study, we asked patients if they perceived un unmet care need in the area of drug misuse, such as cannabis or other addictive substances. However, patients in our study reported no unmet care needs in this area (see Table 2). Various factors may explain this, e.g. patients with cannabis problems were referred to specialized drug treatment centres [4]. Or in the case of current use, patients did not perceive this use as problematic. To provide more information we added the following sentences. 

We added (discussion section, page 14.):

“Another interesting finding is that ADHD patients perceived no unmet care needs for drug misuse or alcohol abuse. This is remarkable because ADHD often co-occurs with substance abuse and dependence (e.g. cannabis misuse) [49-51]. Several factors may explain why children and adolescents expressed no unmet care needs in this area. It may be that actual use was relatively low in our sample because patients with problematic drug misuse or alcohol abuse were referred to specialized drug treatment centres. But it is also possible that patients with problematic alcohol or substance abuse did not perceive their use as a problem.”

Comment 8: 

There are the two general outpatient treatment settings part of the specialized treatment centre? On what organizational level are the general outpatient treatment settings? Are they at a specialized level so that you need to be referred there? Which means that you already have had a previous medical contact. I also think it would be appropriate to describe the presumption that the ACT-group was less psycho-socially well-off here. Highlight that in the beginning of the introduction section that the ACT-group is almost always disadvantaged in the findings, probably as a function of being disadvantaged in several areas. 

Response 8:

If intensive treatment is needed, children and adolescents with ADHD in the Netherlands are referred to a general outpatient clinic or a specialized treatment centre for child and adolescent psychiatric disorders by a general practitioner [4]. During psychiatric treatment, patients treated in a general outpatient setting have (on average) weekly appointments. If more intensive mental health care is required, patients can be referred to youth-ACT. To clarify the treatment settings, we have changed the following sentences.

Original text (background section, page 2.):

 “Most children and adolescents with ADHD in the Netherlands are referred to outpatient clinics for mental health care [7], where mental health professionals focus on reducing their symptoms and improving psychosocial functioning [6, 8]. Common treatments include medication (e.g. stimulants); behavioural therapy and cognitive behavioural therapy; psycho-education, organization and planning-skills training; social skills training; and parental support [9, 10].”

New text:

“If intensive psychiatric treatment is needed, children and adolescents with ADHD in the Netherlands are referred to specialized general outpatient clinics by a general practitioner [7]. Treatment generally focusses on reducing symptoms and improving psychosocial functioning [6, 8]. Common treatments include medication (e.g. stimulants); behavioural therapy and cognitive behavioural therapy; psycho-education, organization and planning-skills training; social skills training; and parental support [9, 10]. If even more intensive mental health care is necessary, patients can be referred to youth Assertive Community Treatment (youth-ACT). ACT is an intensive and outreach-oriented treatment for patients with severe psychiatric and psychosocial problems. Treatment is provided by a multidisciplinary team of mental health care professionals [11-13].”

Comment 9 (abstract, page 2.): 

With respect to the abstract, I think the abstract lacks important information in order for the reader to grasp the study. Ages of participants, number of participants, etc. According to author guidelines (checked 2019-10-11 of PLOS ONE) there is room for another 175 words. What is Youth Assertive Community Treatment? Shouldn't that be explained in the abstract, and that the ACT group is thought to be psychosocially burdened?

Response 9: 

In accordance with the recommendations of the reviewer, we have added a sentence in which we explain what a youth-ACT treatment encompasses and which patient population it serves. Also, we now provide information on the number of patients included in the two samples: general outpatient sample and youth-ACT sample. To provide more information about the youth-ACT setting and number of participants.

Original text (abstract, page 2.):

“Background: Non-attendance, non-compliance, or drop-out from treatment, result in suboptimal treatment of childhood ADHD. These problems may be due to mismatches between needs of patients and treatments provided.”

New text:

“Background: Non-compliance to, or drop-out from treatment for childhood ADHD, result in suboptimal outcome. Non-compliance and drop-out may be due to mismatches between patients' care needs and treatments provided. This study investigated unmet care needs in ADHD patients. Unmet needs were assessed in two different treatment settings (general outpatient setting versus youth-ACT). Youth-ACT treatment is an intensive outreach-oriented treatment for patients with severe psychiatric and psychosocial problems. Comparison of a general outpatient sample with a youth-ACT sample enabled us to assess the influence of severity of psychiatric and psychosocial problems on perceived care needs.”

Original text (abstract, page 2.):

“Methods: Self-reported unmet care needs were assessed among 105 ADHD patients between 6 and 17 years of age in a general outpatient and a youth-ACT setting.”

New text:

“Methods: Self-reported unmet care needs were assessed among 105 ADHD patients between 6 and 17 years of age in a general outpatient (n=52) and a youth-ACT setting (n=53).”

Comment 10 (abstract, page 2.): 

In Conclusions abstract it is stated that “Focusing treatment of ADHD patients on unmet needs that were detected may reduce…”. Perhaps it is a mistake in wording, but it is not said in what way the clinician should detect the unmet needs.

Response 10: 

In the response to comment 8, we clarified that a clinician could detect the patients’ unmet care needs systematically assess using the CANSAS questionnaire during the screening/intake phase. We changed the following sentence.

Original text (abstract, page 2):

“Conclusions: Focusing treatment of ADHD patients on unmet needs that were detected may reduce non-attendance, non-compliance, and drop-out. Systematic assessment of unmet care needs in all ADHD patients may be warranted.” 

New text:

“Conclusions: Focusing treatment of ADHD patients on unmet needs may reduce non-compliance and drop-out. In clinical practice, systematic assessment of unmet care needs in all ADHD patients may be warranted, e.g. using the CANSAS questionnaire during the screening/intake phase.”

Comment 11 (method section, page 4.):

Design section: “This cross-sectional study was conducted between 2015 and 2017 in a specialized treatment centre for child and adolescent psychiatric disorders in the Netherlands.” But there were 2 different settings! It is presented in the next paragraph, but it becomes contradictory. “Setting: Participants were recruited from two general outpatient treatment settings and one youth-ACT team.”

Response 11: 

To clarify the setting where the study was conducted, we changed to the following sentence.

Original text (method section, page 4.):

“This cross-sectional study was conducted between 2015 and 2017 in a specialized treatment centre 

for child and adolescent psychiatric disorders in the Netherlands.”

New text:

“This cross-sectional study was conducted between 2015 and 2017 with patients treated in two general outpatient clinics or a youth-ACT setting, all being part of a large mental health care institution in the Netherlands.”

Comment 12: (method section, participants, Page 5)

 “…. A random sample was selected from the general outpatient population.” Was the investigation of unmet needs done directly when they came on their first visit, or did they have ongoing contact? - If so, for how long / how many visits? When were the interviews done? It is reported that they were done in “…outpatients who had recently started treatment”, but it does matter how long they have been in treatment, doesn’t it? If the patients are still uninformed about their diagnosis after having received a certain amount of treatment is quite different from if it was at their first appointment.

Response 12: 

The feedback from the reviewer highlights the importance of providing additional information about the procedures that were used to determine the care needs of ADHD patients by using the CANSAS. For clarification we added the following sentences to the method section.

We added (method section, page 6.):

“At the outpatient clinics, measurements were conducted on the day of the first appointment (intake). In the youth-ACT setting, measurements after the first (intake) or second appointment. In both settings, measurements for this study took place before patients and parents were informed about results of the clinical assessments.” 

Comment 13: 

The CANSAS is said to have been administered in a face-to-face interview with the patient during the intake procedure. – Were 6-years old interviewed? Did the parents participate? 

Response 13: 

To provide more information regarding the methods that have been used to assess the unmet care needs of ADHD patients, we added the following sentences.

We added (Method section, Measurement instruments, Page 6.):

“For children below the age of 12, the interview was carried out in the presence of the parent. The parent was encouraged to support the child in answering the question if the interviewer felt that the child's answer was unclear. Prior to the interview, parents were instructed not to answer for the child, but to clarify the questions in such a way that the child was able to answer the question from his or her own perspective.”

Comment 14 (method section, page 6): 

Is CANSAS validated for children? 

Response 14: 

Currently, there is no “gold-standard” measure for assessing needs in patients with childhood ADHD. The Camberwell Assessment of Need [5] was used in previous research as a standardized instrument [6]. We changed the following sentences.

Original text (Method section, Measurement instrument, Page 6.):

“To assess a child or adolescent’s met and unmet care needs, the Camberwell Assessment of Need Short Appraisal Schedule (CANSAS) was used as a standardized instrument [37].”

New text: 

“Currently, there is no “gold-standard” for assessing care needs in patients with childhood ADHD. To assess unmet care needs in children and adolescents, the Camberwell Assessment of Need Short Appraisal Schedule (CANSAS) [38] has been used in previous research [27]. The CANSAS was judged as the most appropriate of the available needs assessment instruments, as it is the most widely used needs assessment tool in general mental health services [27].”

Comment 15 (result section, page 13.): 

There are surprisingly small differences (except for GAF) between the two groups as shown in Table 1. Could you comment on that, please? The presumption is that the ACT-group had more severe psychiatric and psychosocial problems, but that can’t be concluded from table 1.

Response 15: 

With the exception of the GAF score, the differences between the two groups were indeed small in terms of age, gender, type of ADHD diagnosis, living situation and country of birth. However, it is also not to be expected that there are differences between the groups on the other variables than the GAF-score. This is because it is in fact poor general functioning of patients, expressed in low GAF-scores, that leads to an intensification of treatment from out-patient care to ACT treatment. 

We performed a t-test independent samples for the variables `age' and `GAF-score', and chi-square tests on `gender’, `type ADHD diagnosis`, `living situation` and `country of birth` (see SPSS output tables below). Comparison between the outpatient sample and the youth-ACT sample showed a significant difference regarding the GAF-score. The average GAF-score of youth-ACT (mean = 46.5) was classified in a lower GAF-score category than the outpatient group (mean = 54.7) indicating that ACT patients are more likely to experience severe psychiatric symptoms (e.g. suicidal thoughts, severe obsessive rituals, frequent shoplifting) and severe limitations in social and/or school functioning (e.g. no friends, inability to practice a hobby/sport, and/or none-attendance of school).

To provide complete information about the tests we performed (including on age and GAF scores), we changed the main text of the manuscript as follows. 

Original text (method section, data-analysis, page 7): 

“Subgroup differences were analysed using the Chi-square test with Yates continuity correction (= χ2 -test), or, if the number in at least one of the cells was lower than 5, with the Fisher Exact test [42].” 

Nex text:

“Subgroup differences were analysed using the t-test for continuous variables, chi-square test with Yates continuity correction (= χ2 -test), or, if the expected number in at least one of the cells was smaller than 5, with the Fisher Exact test [42].” 

We added (result section, page 7): 

“Table 1 shows demographic characteristics of our two samples. Patients in the outpatient sample had significantly higher GAF-scores than those in the youth-ACT sample (mean = 54.7, sd = 5.5 vs. mean = 46.5, sd = 8.3). There were no significant differences between the outpatient sample and youth-ACT sample regarding age, gender, country of birth, type of ADHD diagnosis, and living situation.” 

Further, we changed the content of table 1 (see manuscript) by adding an extra column. In this extra column we now present the differences between the two settings analyzed using t-test for continuous variables, or Chi-square test with Yates continuity correction (χ2 -test) for categorical variables. As an alternative for the Chi-square test, the Fisher's Exact test was computed if the number in at least one of the cells of the categorical variable was lower than 5. 

We added (discussion section, comparing outpatient clinics with youth-ACT, page 16)

“The differences between the two settings regarding unmet needs cloud not be explained by age, gender, type of ADHD diagnosis, living situation or country of birth. However, comparison between the two treatment settings showed a significant difference regarding the GAF-score, indicating that ACT patients had more problems in daily functioning [2].”

Comment 16 (result section, page 8): 

The fact that patients who have come to specialized CAP care to such an extent (60%) considered 

themselves uninformed about diagnosis is remarkable, a major finding that should affect care routines and this should be mentioned as a first finding and emphasized in the discussion.

Response 16: 

We agree with the reviewer that being informed about diagnosis and treatment is an important issue. 

Therefore, we stated in the result section: “As Table 2 shows, mental health problems were the most self-reported unmet care need of children and adolescents with ADHD: 61% reported an unmet need in this area (outpatient sample 66.0%; youth-ACT sample 57.7%; n.s.). The second most self-reported unmet care need—which was reported by 47.6% of all included patients—concerned information on diagnosis and treatment. There was a significant difference between the groups: 60.4% of the outpatient sample vs. 34.6% of the youth-ACT sample (p < .05).”

To emphasize that 60% patients who have come to a general outpatient care setting considered themselves uninformed about diagnosis and/or treatment, we added the following sentences in the discussion section.

We added (discussion section, page 15-16.):

“For clinical practice, our findings suggest that many patients consider themselves uninformed about assigned diagnoses (60%) in the general outpatient group, and one third in the youth-ACT group. In both settings, clinicians should pay close attention to providing in information about diagnosis and treatment.”

Comment 17 (discussion section, page 18.):

”… Overall, ADHD patients reported substantial levels of unmet needs in several areas…” The authors argue that they found substantial levels of unmet needs. But how many percent indicates high or low? Are there any studies to compare with? 

Response 17: 

Unfortunately, to our knowledge, criteria that indicate whether the percentages of reported unmet care

needs are high or low are not available. To our knowledge, we performed the first study that provide a detailed insight into the perceived unmet care needs of children and adolescents with ADHD. However, our study can be compared with the study conducted by Eklund et al. [6] that partly included adolescents with ADHD. Therefore, we changed the following sentence.

Original text (discussion section, page 14.): 

“Overall, ADHD patients reported substantial levels of unmet needs in several areas.”

New text:

“Compared to young adults, in whom unmet care needs in various areas were found to a frequency of up to thirty percent [27], children and adolescents with ADHD reported levels of unmet needs up to sixty percent.”

Comment 18: 

You indicate on page 10 that the study was driven by two a priori hypotheses. Please comment on if your findings support those hypotheses.

Response 18: 

We described in the introduction: “On the basis of the literature, we had two a priori hypotheses: (1) that ADHD patients treated in the youth-ACT setting would experience more unmet care needs than those treated in a general outpatient care setting [11, 13, 33]; and (2) that the greatest differences between patients in the two settings would involve participation in the community, with more recipients of youth-ACT perceiving that their care needs were not being met [2, 34, 35].” In line with the reviewer's suggestion, we have added the following sentences to the manuscript to comment on the hypotheses we mentioned earlier in the introduction. 

Original text (discussion section, page 15.):

“In contrast with our expectations, patients at outpatient clinics reported more unmet needs in the domains of physical and mental functions and performance of daily activities than those receiving youth-ACT, a more intensive treatment. In the domain of physical and mental functions, outpatients with ADHD were more likely than those in the youth-ACT sample to perceive unmet needs with respect to information on treatment and diagnosis.”

New text:

“In line with our a priori hypotheses, ADHD patients from the youth-ACT setting reported significantly more unmet care needs than those treated in the general outpatient care setting. The notable exception, in the domain of physical and mental functions, was that outpatients with ADHD were more likely than those in the youth-ACT sample to perceive unmet needs with respect to information on diagnosis and treatment.”

Original text (discussion section, page 15.):

“We should also draw attention to the high score of unmet care needs related to the domain of participation in the community in the youth-ACT sample. As youth-ACT focuses specifically on enhancing patients' societal functioning, this score indicates that most of these patients had been referred to the appropriate treatment setting.”

New text:

“We should also draw attention to the high level of unmet care needs related to participation in the community in the youth-ACT sample. This finding is in line with our a priori hypotheses. The largest difference between patients from the two settings involved participation in the community. Recipients of youth-ACT perceived more unmet care needs in this area. As youth-ACT focuses specifically on enhancing patients' societal functioning, this score indicates that most of these patients had been referred to the appropriate treatment setting.”

Comment 19: 

”For the youth-ACT sample, we included all patients who were referred to this treatment setting during the inclusion period. These ACT-patients all had received prior general outpatient treatment.” Does this mean that they had had contact with general CAP outpatient treatment? Or with primary care?

Response 19: 

Yes, all ACT patients previously received outpatient care. All included clients were initially referred by the general practitioner to an outpatients clinic. 

Comment 20: 

As I mentioned above the Background section is well written. However reference number 1 is a bit old. This one is more updated: Polanczyk, G. V., Salum, G. A., Sugaya, L. S., Caye, A., & Rohde, L. (2015). Annual research review: A meta-analysis of the worldwide prevalence of mental disorders in children and adolescents. J Child Psychol Psychiatry, 56(3), 345-365. doi:10.1111/jcpp.12381.

Response 20:

 As recommended, we have changed reference number 1 [7]. We now refer to a more recent meta-analysis of the worldwide prevalence of mental disorders in children and adolescents [8].

Original text (references, page 18.):

“1. Belfer M. Child and adolescent mental disorders: the magnitude of the problem across the globe. J Child Psychol Psychiatry. 2008;49(3):226-36. https://doi.org/10.1111/j.1469-7610.2007.01855.x PMID: 18221350.”

New text:

“1. Polanczyk G, Salum G, Sugaya L, Caye A., Rohde L. Annual research review: a meta-analysis of the worldwide prevalence of mental disorders in children and adolescents. J Child Psychol Psychiatry. 2015;56(3):345-65. https://doi.org/10.1111/jcpp.12381 PMID: 25649325”

Comment 21: 

The authors argue at the bottom of page 9 that the perspective of patients with ADHD is not much investigated. However, there is at least one study: Emilsson et al. Beliefs regarding medication and side effects influence treatment adherence in adolescents with attention deficit hyperactivity disorder. Eur Child Adolesc Psychiatry 2017;26:559-571.

Response 21: 

The article “Beliefs regarding medication and side effects influence treatment adherence in adolescents with attention deficit hyperactivity disorder” [9], that is mentioned by the reviewer, describes a study that investigated the perceptions /beliefs of adolescents with ADHD regarding their illness, medication and side-effects. Although this article was not focussed on identification of unmet care needs, research by Emilsson et al. showed that beliefs of adolescents with ADHD about their illness and medication side effects influenced treatment adherence. This shows that it is important to obtain information of patients themselves regarding their own view on their situation. Therefore, it seems appropriate to refer to this article in our introduction. We added the following sentences.

We added (background, page 4):

“Moreover, insight into the perception of ADHD patients may help to enhance their adherence to treatment [31].”

We added (references, page 19):

“31. Emilsson M, Gustafsson P, Öhnström G, Marteinsdottir I. Beliefs regarding medication and side effects influence treatment adherence in adolescents with attention deficit hyperactivity disorder. Eur Child Adolesc Psychiatry. 2017;26(5);559-71. https://doi.org/10.1007/s.00787-016-0919-1 PMID: 2784823”

Comment 22: (method section, measurement instruments, page 6.) 

Was MINI-KID made as part of the research investigation or was it routine measure for clinical intake? Who did the MINI-KID? Did you have any supplementary diagnostic measures like questionnaires from parents, teachers, etc. ?

Response 22: 

In the two general outpatient clinics, the MINI-KID was used for research purposes. But in the 

youth-ACT setting, the MINI-KID was part of the standard routine measurement instruments. 

In both settings the MINI-KID was administered in a standardized manner by case managers who were trained in using this instrument. 

References

[1] R. Patel, P. Patel, K. Shah, M. Kaur, Z. Mansuri and R. Makani, "Is cannabis use associated with the worst inpatient outcomes in attention deficit hyperactivity disorder adolescents?," Cureus, vol. 10, no. 1, pp. 1-10, 2018. 

[2] I. Elkins, G. Saunders, S. Malone, M. Keyes, M. McGue and W. Lacono, "Associations between childhood ADHD, gender, and adolescent alcohol and marijuana involvement: a causally informative design," Drug Alcohol Depend, vol. 184, pp. 33-41, 2018. 

[3] S. Lee, K. Humphreys, R. Flory and N. Halfon, "Prospective association of childhood attention-deficit/hyperactivity disorder (ADHD) and substance abuse/dependence: a meta-analytic review," Clin Psychol Rev , vol. 31, no. 3, pp. 328-341, 2011. 

[4] GGZ Nederland, "Sectorrapport ggz 2013," GGZ Nederland, Amersfoort, 2013.

[5] M. Phelan, M. Slade and G. Thornicroft, "The camberwell assessment of need (CANSAS): the validity and reliability of an instrument to assess the needs of people with severe mental illness," Br J Psychiatry, vol. 167, no. 5, pp. 589-95, 1995. 

[6] H. Eklund, J. Findon, T. Cadman, H. Hayward, D. Murphy, P. Asherson, K. Glaser and K. Xenitidis, "Needs of adolescents and young adults with neurodevelopental disorders: comparison of young people and parents perspectives," J Autism Dev Disord, vol. 48, no. 1, pp. 83-91, 2018. 

[7] M. Belfer, "Child and adolescent mental disorders: the magnitude of the problems across the globe.," J. Child Psychol Psychiatry, vol. 49, no. 3, pp. 226-36, 2008. 

[8] G. Polanczyk, G. Salum, L. Sugaya, A. Caye and L. Rohde, "Annual research review: a meta-analysis of the worldwide prevelence of mental disorders in children and adolescents," J Child Psychol Psychiatry, vol. 56, no. 3, pp. 345-65, 2015. 

[9] M. Emilsson, P. Gustafsson, G. Öhnström and I. Marteinsdottir, "Beliefs regarding medication and side effects influence treatment adherence in adolescents with attention deficit hyperactivity disorder.," Eur Child Adolesc Psychiatry, vol. 26, no. 5, pp. 559-71, 2017. 

[10] A. Lasalvia, M. Rugerri, M. Mazzi and R. Dall' Agnola, "The perception of need for care in staff and patients in community-based mental health services. The South-Verona outcome project 3," Acta Psychiatr Scand, vol. 102, no. 5, pp. 366-75, 2000.

---

## [Decision Letter · Decision Letter 1]

7 Jan 2020

Unmet Care Needs of Children with ADHD

PONE-D-19-19638R1

Dear Dr. Vijverberg,

We are pleased to inform you that your manuscript has been judged scientifically suitable for publication and will be formally accepted for publication once it complies with all outstanding technical requirements.

With kind regards,

Michelle Tye, Ph.D.

Academic Editor

PLOS ONE

Additional Editor Comments (optional):

The authors have done a thorough and excellent job of addressing all reviewer comments point-by-point. Minor editorial errors can be addressed prior to the production stage. 

Reviewers' comments:

Reviewer's Responses to Questions

**Comments to the Author**

1. If the authors have adequately addressed your comments raised in a previous round of review and you feel that this manuscript is now acceptable for publication, you may indicate that here to bypass the “Comments to the Author” section, enter your conflict of interest statement in the “Confidential to Editor” section, and submit your "Accept" recommendation.

Reviewer #2: All comments have been addressed

2. Is the manuscript technically sound, and do the data support the conclusions?

Reviewer #2: Yes

3. Has the statistical analysis been performed appropriately and rigorously? 

Reviewer #2: Yes

4. Have the authors made all data underlying the findings in their manuscript fully available?

Reviewer #2: Yes

5. Is the manuscript presented in an intelligible fashion and written in standard English?

Reviewer #2: Yes

6. Review Comments to the Author

Reviewer #2: The authors have responded in an adequate manner to all my previous suggestions. However, on page 15 line 8 of the revised manuscript there is a miss-spelling, "cloud" should be "could".

7. PLOS authors have the option to publish the peer review history of their article (what does this mean?). If published, this will include your full peer review and any attached files.

Reviewer #2: Yes: Per A Gustafsson

---

## [Editor Report · Acceptance letter]

9 Jan 2020

PONE-D-19-19638R1 

Unmet Care Needs of Children with ADHD 

Dear Dr. Vijverberg:

I am pleased to inform you that your manuscript has been deemed suitable for publication in PLOS ONE. Congratulations! Your manuscript is now with our production department. 

With kind regards,

on behalf of

Dr. Michelle Tye 

Academic Editor

PLOS ONE